Long-term outcomes of repaired and unrepaired truncus arteriosus: 20-year, single-center experience in Thailand

Dangrungroj Ekkachai 1
Vijarnsorn Chodchanok cvijarnsorn@yahoo.com 1
Chanthong Prakul 1
Chungsomprasong Paweena 1
Kanjanauthai Supaluck 1
Durongpisitkul Kritvikrom 1
Soongswang Jarupim 1
Tantiwongkosri Kriangkrai 2
Subtaweesin Thaworn 2
Sriyoschati Somchai 2
1 Department of Pediatrics, Faculty of Medicine Siriraj Hospital, Mahidol University , Bangkok , Thailand
2 Department of Surgery, Faculty of Medicine Siriraj Hospital, Mahidol University , Bangkok , Thailand
Zhan Cheng
Electronic publication date: 2020 May 12
Publication date: 2020
Volume: 8
Electronic Location ID: e9148
Received 2019 Sep 16; Accepted 2020 Apr 17
Copyright: ©2020 Dangrungroj et al.
Copyright year: 2020
Copyright holder: Dangrungroj et al.
License: This is an open access article distributed under the terms of the Creative Commons Attribution License, which permits unrestricted use, distribution, reproduction and adaptation in any medium and for any purpose provided that it is properly attributed. For attribution, the original author(s), title, publication source (PeerJ) and either DOI or URL of the article must be cited.
License URL: https://creativecommons.org/licenses/by/4.0/

Keywords: Truncus arteriosus, Survival, Mortality risk, Congenital heart defect

Funding: The authors received no funding for this work.

==============================
Background

Truncus arteriosus (TA) is a complex congenital heart disease that carries morbidities in the first year of life. Previous authors have reported an operative mortality of 50%. In this report, we aim to report on the survival of patients with TA in our medical center in the recent era.

Methods

A retrospective review of all patients diagnosed with TA in Siriraj Hospital, Thailand from August 1995 to March 2018 was performed. Patients with single ventricle, hemiTA were excluded. The characteristics and outcomes of repaired and unrepaired TA patients with a known recent functional status in 2018 were reviewed. Operative mortality risks were analyzed using a multivariate model.

Results

A total of 74 patients (median age at referral: 70 days) were included in the cohort. One-third of the patients had associated anomalies including DiGeorge syndrome (13.5%). Anatomical repair was not performed in 22 patients (29.7%). The median age at time of repair for the 52 patients was 133 days (range: 22 days to 16.7 years). Complex TA was 10%. Early mortality occurred in 16 patients (30.8%). Five patients (9.6%) had late deaths at 0.3–1.2 years. Significant mortality risk was weight at time of operation <4 kg (HR 3.05, 95% CI [1.05–8.74], p-value 0.041). Of the 31 operation survivors, 17 required re-intervention within 0.4–11.4 years. Eight patients had reoperation at 8.7 years (range: 2.7–14.6 years) post-repair. Freedom from reoperation was 93%, 70.4%, and 31%, at 5, 10, and 15 years, respectively. All late survivors were in functional class I–II. Of the 22 unrepaired TA patients, 11 patients (50%) died (median age: 13.6 years; range: 14 days–32.8 years). Survival of unrepaired TA patients was 68.2%, 68.2%, and 56.8, at 5, 10, and 15 years of age, respectively. At the end of study, 11 survivors of TA with palliative treatment had a recent mean oxygen saturation value of 84.1 ± 4.8% and a mean weight for height of 81.4 ± 12.7%, which were significantly lower than those of 31 late-survivors who had undergone anatomical repair.

Conclusion

Contemporary survival rates of patients with TA following operation in the center has been gradually improved over time. Most of the operative mortality occurs in the early postoperative period. Compared to patients with TA who had palliative treatment, operative survivors have a better functional status even though they carry a risk for re-intervention.

Introduction

Truncus arteriosus (TA) is a rare, complex congenital heart disease characterized by a single great artery supplying both systemic and pulmonary circulations, as the so called common arterial trunk (Collett & Edwards, 1949). Based on Collett and Edwards’ classification (Collett & Edwards, 1949; Jacobs, 2000), TA was classified into four anatomical types originating from the pulmonary artery. However, type IV currently corresponds to pulmonary atresia with ventricular septal defect, which is not precise TA nomenclature. Clinically, TA leads to cyanosis because of the mixing of deoxygenated and oxygenated blood at the common arterial trunk. Heart failure is inevitable when the pulmonary vascular resistance physiologically declines in infancy. Truncal valve regurgitation and aortic arch anomaly often aggravate heart failure symptoms and mortality (Jacobs, 2000). If patients were left untreated, pulmonary vascular adaptive mechanisms would lead to rapid pulmonary arterial hypertension and subsequently to Eisenmenger syndrome (ES) (Marcelletti, McGoon & Mair, 1976).

Anatomical repair of TA was reported in 1967 by Rastelli, Titus & McGoon (1967) and surgical techniques have progressed steadily since then  (Mavroudis, Jonas & Bove, 2015). The timing of the operation tends to be an early repair in neonates or during infancy within the first three months of age to reduce the risk of pulmonary hypertension (Brown et al., 2001; Hanley et al., 1993). In any case, the surgical procedure is widely known to carry a significant risk of mortality and the operation requires meticulous postoperative management. According to the Society of Thoracic Surgeons Congenital Heart Surgery database (STS-CHSD) from 2000–2009, TA repair with truncal valve surgery had a significantly higher rate of mortality than without truncal valve surgery (30% vs. 11%) (Russell et al., 2012). Furthermore, in a recent multicenter cohort study in the US, 20% of the children who underwent simple TA repair were reported to experience major adverse cardiac events postoperatively (Mastropietro et al., 2019). Currently, complex TA surgery is categorized as a level-4 procedure according to the Society of Thoracic Surgeons—European Association for Cardio-Thoracic Surgery (Russell et al., 2012; Mastropietro et al., 2019).

Siriraj Hospital is one of referral cardiac centers that performs surgical corrections and provides postoperative care for children with TA in Thailand. In a report of early cases prior to 2004, Loahaprasitiporn and colleagues found an operative mortality of 50% following corrective surgery, within 6 weeks to 6 months of age (Laohaprasitiporn et al., 2008). Medically conservative treatments were provided to patients who did not meet operability criteria, such as late referral with ES. Currently, some survivors in both repaired and unrepaired groups reach adulthood. In the present study, we aim to evaluate: (1) the survival rates of patients who were diagnosed with TA, and either repaired or unrepaired TA, at Siriraj Hospital; and (2) operative mortality risks for the past two decades.

Materials and Methods

This study was a single-centered, observational study using a hospital database from a large referral cardiac center in Thailand. Following approval from the Siriraj Institutional Review Board Faculty of Medicine, Siriraj Hospital, Mahidol University (COA no. Si 379/2017), all consecutive patients who had a confirmed diagnosis of TA type I–III (Collett and Edward) (Collett & Edwards, 1949) by echocardiography between August 1, 1995 and March 31, 2018 in the center were retrospectively reviewed. Patients with single ventricle, hemiTA, or TA type IV, or patients who had undergone an operation from another hospital were excluded. The requirement for informed consent from patients was waived with the approval of the Siriraj Institutional Review Board Faculty of Medicine, Siriraj Hospital, Mahidol University. Demographic data was collected for age at referral, diagnosis, gender, weight at referral, oxygen saturation, associated anomalies, presence of DiGeorge syndrome, initial presentation, and cardiac findings including type of TA (I, II, III), truncal valve leaflets, degree of truncal valve regurgitation, presence of aortic arch interruption, pulmonary artery stenosis, arch sidedness and coronary abnormality. Patient management was recorded, including surgical interventions, postoperative complications, and clinical outcomes including weight, height, functional class, oxygen saturation, and mortality following diagnosis and at the most recent follow-up at the end of 2018. Patients who were lost to a follow-up or their latest functional status was unknown in 2018 were excluded from the study cohort. Early mortality following total correction for TA was defined as 30-day mortality including patients who died after 30 days without being discharged from the hospital when admitted for the operation. Patient information was anonymized and de-identified prior to the analysis.

Statistical analysis

The patients’ baseline characteristics and outcomes were summarized using descriptive statistics. Normally distributed data was presented as mean  ± SD, while the median (with range) was used where the distribution of data was not normal. Categorical data was represented as a number and a percentage (%).The comparison of continuous variables between groups was assessed using an unpaired T test for normally distributed data and a Wilcoxon rank-sum test for non-normally distributed data. Differences between the categorical data were assessed with a Chi-square or Fisher’s exact test. Patients were classified into two groups; those who underwent anatomical repair for TA and those who had palliative treatment. Cumulative survival, from date of birth and date of operation to the endpoint, was calculated using the Kaplan-Meire analysis with log-rank test. The association between baseline characteristics and postoperative mortality was evaluated with multivariate analysis. A p-value < 0.05 was considered to be statistically significant. Statistical analyses were performed with SPSS 20.0 for Windows (SPSS Inc., Chicago, IL, USA).

Results

Demographics

A total of 74 patients with TA (45% were male) was included in the study cohort. Median age of referral and diagnosis at the center was 70 days of age (range: 0–25.9 years). The patients’ demographics are summarized in Table 1. Almost one-third of the patient population had an associated anomaly such as VACTREL association, anorectal malformation, tracheoesophageal fistula, micropthalmia, hypospadia, including DiGeorge syndrome (13.5%). The most common type of TA was type I. Six patients had pulmonary artery stenosis (8.1%), five patients had moderate or severe truncal valve regurgitation (6.7%), four patients had partial anomalous pulmonary venous return (5.4%), and two patients had interrupted aortic arch (2.7%). Following the identification of operability and obtaining consent from the patients’ parents, 52 patients underwent TA anatomical repair and 22 patients did not. Two patients in the unrepaired TA group underwent palliative pulmonary artery banding (PAB); one was unable to perform further anatomical repair for TA due to underlying Jacobsen syndrome with severe thrombocytopenia while the other died after repair interrupted the aortic arch concurrent with PAB. From observations, the initial oxygen saturation in the repaired TA group was significantly higher than that of the patients who received palliative treatment. The flow of the study is shown in Fig. 1.

Table 1 Patients’ characteristics (n = 74).

Variable	All patients	Repaired TA	Unrepaired TA	p-value	
	(n = 74)	(n = 52)	(n = 22)		
Male gender	34 (45.9%)	25 (48.1%)	9 (40.9%)	0.57	
Age at suspicion of TA at primary hospital (years)	0.05 (0–25.91)	0.05 (0–2.01)	0.07 (0–25.91)	0.79	
Age at referral and diagnosis of TA at the center (years)	0.19 (0–25.91)	0.16 (0–2.01)	1.85 (0–25.91)	0.07	
Body weight at referral (kg)	3.67 (1.50–48.00)	3.38 (1.50–9.40)	8.05 (2.30–48.00)	0.05	
Oxygen saturation, %	88.88 ± 5.86	90.33 ± 4.68	85.45 ± 6.96	0.005*	
Cardiothoracic ratio on chest radiography	0.62 ± 0.05	0.62 ± 0.04	0.63 ± 0.06	0.42	
Prenatal diagnosis	4 (5.4%)	3 (5.7%)	1 (4.5%)	1.00	
Associated anomalies	24 (32.4%)	18 (34.6%)	6 (27.3%)	0.53	
- DiGeorge syndrome	10 (13.5%)	8 (15.4%)	2 (9.1%)	0.47	
TA type (Collett and Edwards)					
- Type I	54 (72.9%)	37 (71.1%)	17 (77.3%)	0.58	
- Type II	17 (23.0%)	13 (25%)	4 (18.2%)		
- Type III	3 (4.1%)	2 (3.9%)	1 (4.5%)		
Truncal valve leaflets					
- Bicuspid	14 (19.0%)	9 (17.3%)	5 (22.7%)	0.84	
- Tricuspid	40 (54.0%)	31 (59.6%)	9 (40.9%)		
- Quardricuspid	20 (27.0%)	12 (23.1%)	8 (36.4%)		
Presence of moderate and severe truncal valve regurgitation	5 (6.7%)	4 (7.7%)	1 (4.5%)	1.00	
Presence of pulmonary artery stenosis	6 (8.1%)	5 (9.6%)	1 (4.5%)	0.66	
Presence of interrupted aortic arch	2 (2.7%)	1 (1.9%)	1 (4.5%)	0.51	
Presence of right side aortic arch	22 (29.7%)	17 (32.7%)	5 (22.7%)	0.35	
Presence of partial anomalous pulmonary venous return	4 (5.4%)	4 (7.7%)	0	0.31	
Left ventricular ejection fraction (%)	66.99 ± 8.90	68.53 ± 8.09	62.80 ± 9.86	0.015*	
Notes.

Data represented by median (range), mean ± SD and number (% within column).

Statistically significant at p-value < 0.05 by Chi-square or Fishers exact test and independent T test (for normally distributed data) or Wilcoxon rank-sum test (for non-normally distributed data).

TA truncus arteriosus

Surgical correction and mortality

Table 2 shows patient demographics, operative data, and early postoperative course in 52 repaired TA patients. Anatomical repair of TA has been performed at the medical center since 1999. Of the 52 patients, 19 underwent total repair in 1999–2006, representing various age groups at the time of operation: neonate (n = 2), aged 1–6 months old (n = 31), aged 6 months–1 year old (n = 13), age more than 1 year old (n = 6). Three patients underwent PAB prior to a total correction. The right ventricular to pulmonary artery valve conduit diameter varied from 12–25 mm, depending on the patient’s size, with z-score of 2.3 ± 0.9. The types of conduit are shown in Table 2. One patient in the cohort underwent direct anastomosis from right ventricle to pulmonary artery. Repair of complex TA, which is defined as TA with significant pathology of truncal valve and aortic arch (Russell et al., 2012; Mastropietro et al., 2019), accounted for 5 of 52 (9.6%) of the cohort). Among four patients who underwent concurrent truncal valve repair, three died postoperatively. The patient who survived (13.9 years postoperatively) had progressive aortic valve (native truncal valve) and conduit regurgitation, and the patient is currently on a list for redo surgery. One patient who underwent concomitant aortic arch repair for interrupted aortic arch died early post-operation due to pulmonary hypertensive crisis. A total of 16 early deaths (30.8%) occurred with a median postoperative time of 3 days (range: 0–40 days). Most of the patients with early mortality died from low cardiac output syndrome (LCOS), related to pulmonary hypertensive crisis or myocardial ischemia, followed by sepsis and multi-organ dysfunction (Table 3). All early survivors (n = 36) were complete in a follow up with a median duration of 5.7 years (range: 0.46–19.8 years). Late mortality was reported in five patients (9.6%) at a median duration of seven months after operation (range: 5.5 months–1.2 years), related to infection and possible persistent pulmonary hypertension postoperatively (Table 3). The postoperative survival of patients with repaired TA (n = 52) was 63.5% at 1-year and steady at 59.5% after 2 years following the operation. Consequently, survival rates of early survivors (n = 36) at 1, 5, 10, and 15 years following repair were 91.7%, 85.9%, 85.9%, and 85.9%, respectively. A univariate analysis of 17 variables revealed 2 potential risk factors of mortality following repaired TA: weight at operation <4 kg and preoperative major infection within 1 month (Table 4). The independent risk of postoperative mortality, which was identified by multivariate analysis was only weight at operation <4 kg (HR 2.71, 95% CI [1.05–6.95]; p-value 0.039) (Table 4). With regards to early death postoperatively, weight at operation <4 kg was found to be an associated factor by univariate and multivariate analyses (HR 3.05, 95% CI [1.05–8.74], p-value 0.041). In patients with late mortality, preoperative major infection within one month and pulmonary hypertensive crisis in the operating room requiring inhaled nitric oxide were shown to be mortality risks by the univariate analysis. Nevertheless, only pulmonary hypertensive crisis in the operating room requiring inhaled nitric oxide was found to be an independent risk factor for late death by the multivariate analysis (hazard ratio 6.59, 95% CI [1.02–42.7], p-value 0.048).

Figure 1 Flow of study.

Reoperation and reintervention

Thirty-one late survivors following total correction were reported surviving until the end of 2018 and were having regular follow-ups at the medical center. The median postoperative follow-up was 6.4 years (range: 1.0–19.8 years). Seventeen patients (54.8%) required either catheter intervention or reoperation for conduit change at a median time of 3.1 years post-operation (range: 0.4–11.4 years). The range in number of catheter interventions was 0–4 times per patient. Freedom from re-intervention was reported to be 93.3%, 48.3%, 43.9%, and 35% at 1, 5, 10, and 15 years post-repair, respectively. Reoperation for conduit replacement had been performed in 8 patients (25.8%) at the median time of 8.7 years (range: 2.7–14.6 years) after their corrective surgery. Freedom from reoperation in survivors with repaired TA was 100%, 93%, 70.4%, and 31% at 1, 5, 10, and 15 years postoperation, respectively.

TA with palliative treatment

Twenty-two patients in the cohort had not undergone anatomical repair for TA. Cardiac catheterization, which was performed in 6 patients showed mean pulmonary arterial pressure of 70.8 ± 14.0 mmHg, baseline pulmonary vascular resistance index of 20.1 ± 14.11 WU m2 and post-pulmonary vasodilator testing pulmonary vascular resistance index of 6.9 ± 4.9 WU m2. Some patients who were deemed to be receiving a conservative treatment were referred back to the primary hospital. All 22 patients in this study had been in recent contact and had a known clinical status in 2018. Since their most recent visit (median age: 13.6 years; range: 14 days–32.8 years), 11 patients (50%) had died. Seven patients had died in infancy (less than 1.5 years of age), and the primary cause of death was heart failure, precipitated by infections such as sepsis or pneumonia. Four patients who died (13–32.8 years of age) had late referral and diagnoses of ES. Of the four patients, two died from pneumonia and two patients had sudden death, which was likely related to adverse cardiac events. The survival rates of the TA with palliative treatment were 72.7%, 68.2%, 68.2%, and 56.8, at 1, 5, 10, and 15 years of age, respectively. In comparing the survival ages of 52 patients with TA repair, 63.5% were surviving at 1-year and 59.6% were steady at 5-years of age, with no statistical differences. The survival of 36 early-survivors following TA repair was significantly higher than that of 22 patients with palliative treatment (Log rank p 0.03). At the end of study cohort in 2018, 11 survivors of TA with palliative treatment had a recent mean oxygen saturation value of 84.1 ± 4.8% and a mean weight for height of 81.4 ± 12.7%, which were significantly lower than those of 31 late-survivors who had undergone anatomical repair.

Pulmonary arterial hypertension (PAH) therapy

Of the 52 patients who underwent surgical correction, 47 patients (90.3%) received postoperative pulmonary vasodilators including inhaled nitric oxide, iloprost, or oral forms such as sildenafil, bosentan or beraprost sodium. In five patients who had not been given pulmonary vasodilators, four underwent total correction prior to 2000, when inhaled nitric oxide was unavailable in the theatre and intensive care unit. Sixteen early deaths (30.8%) occurred with a median postoperative time of three days (range: 0–40 days), related to pulmonary hypertensive crisis or myocardial ischemia, followed by sepsis and multi-organ dysfunction. Twenty-eight patients were prescribed oral pulmonary vasodilators when they were discharged home after their operation. Monotherapy was given: beraprost in 3 patients and sildenafil in 25 patients. The median time of post-operative pulmonary arterial hypertension therapy was 6.9 months (range 1–109 months). Pre-treatment targeted therapy was used in two patients who underwent total correction at age 14 and 16 years, because of late referrals and the decision of parents. They were placed on sildenafil (25–50 mg oral three times/day) prior to the cardiac catheterization in the referral center. Interestingly, both patients had baseline oxygen saturation in room air of 90–94%. Cardiac catheterization preoperatively showed elevated pulmonary artery pressure (PAP) in both patients. The first patient had a mean PAP (mPAP) of 80 mmHg, Qp:Qs of 1.36, and pulmonary vascular resistance (PVR) index of 13.6 WU m2 in room air and mPAP 72 mmHg; the PVR index declined to 0.9 WU m2 after inhaled nitric oxide 20 PPM testing. The second patient had a mPAP of 63 mmHg, Qp:Qs of 6, PVR of 3.4 WU, and PVR index of 3.9 WU m2 in room air. Following surgical repair, these two patients continued with the pulmonary vasodilator (oral sildenafil, for 48.6 and 57.9 months following operation, respectively). Repeat cardiac catheterizations at two years post repair were performed, with a decreased mean PAP to 30 mmHg in the first patient and 28 mmHg in the second patient.

Table 2 Demographics, operative data and early postoperative course of patients who underwent truncus arteriosus repair (n = 52).

Variable	All TA repair (n = 52)	Mortality (n = 21)	Survivors in 2018 (n = 31)	p-value	
Male gender	25 (48.1%)	9 (42.9%)	16 (51.6%)	0.53	
Prenatal diagnosis	3 (5.7%)	2 (9.5%)	1 (3.2%)	0.55	
Age at surgery (days)	133 (22–6111)	88 (22–319)	189 (48–6111)	0.11	
Weight at total repair (kg)	4.2 (2.2–38.0)	3.6 (2.2–6.5)	5.1 (2.9–38.0)	0.06	
Associated anomalies	18 (34.6%)	11 (52.4%)	7 (22.5%)	0.027*	
- DiGeorge syndrome	8 (15.4%)	4 (19.0%)	4 (12.9%)	0.54	
TA type I (Collett and Edward)	37 (71.1%)	13 (61.9%)	24 (77.4%)	0.23	
Presence of moderate and severe truncal valve regurgitation	4 (7.7%)	3 (14.3%)	1 (3.2%)	0.29	
Presence of pulmonary artery stenosis	5 (9.6%)	3 (14.3%)	2 (6.5%)	0.38	
Presence of interrupted aortic arch	1 (1.9%)	1 (4.7%)	0	0.40	
Presence of right side aortic arch	17 (32.7%)	7 (33.3%)	10 (32.6%)	0.93	
Presence of partial anomalous pulmonary venous return	4 (7.7%)	0	4 (12.9%)	0.13	
Presence of coronary abnormalities	8 (15.4%)	4 (19.0%)	4 (12.9%)	0.54	
Pulmonary artery banding prior to repair	3 (5.8%)	1 (4.8%)	2 (6.5%)	1.00	
Preoperative major infection within 1 month	10 (19.2%)	7 (33.3%)	3 (9.7%)	0.03*	
Preoperative being on mechanical ventilator	2 (3.8%)	1 (4.8%)	1 (3.2%)	1.00	
Operative data					
Surgical era 1999-2006	15 (28.8%)	7 (33.3%)	8 (25.8%)	0.55	
Cardiopulmonary bypass time (min)	169.6 ± 43.8	181.3 ± 51.1	161.7 ± 36.9	0.07	
Aortic clamp time (min)	107.2 ± 29.6	105.8 ± 26.6	108.1 ± 31.9	0.79	
Truncal valve repaired	4 (7.7%)	3 (14.3%)	1 (3.2%)	0.14	
Aortic arch repaired	1 (1.9%)	1 (4.8%)	0	0.22	
Conduit size (mm)	14.2 ± 3.1	12.9 ± 2.9	15.0 ± 2.9	0.01*	
Conduit size z-score	2.3 ± 0.9	2.3 ± 0.8	2.3 ± 1.0	0.89	
Type of conduit				0.18	
- Aortic homograft	5 (9.6%)	0	5 (16.1%)		
- Pulmonary homograft	10 (19.2%)	3 (14.3%)	7 (22.6%)		
- Handcock/Carpentier–Edwards valved conduit	12 (23.1%)	5 (23.8%)	7 (22.6%)		
- Contegra bovine jugular valved conduit	24 (46.2%)	12 (57.1%)	12 (38.7%)		
-Direct anastomosis with monocusp	1 (1.9%)	1 (4.8%)	0		
Intraoperative usage of inhaled nitric oxide	9 (17.3%)	6 (28.5%)	3 (9.6%	0.07	
Early postoperative course					
Postoperative ECMO	4 (7.7%)	4 (19.0%)	0	0.02*	
Postoperative usage of inhaled nitric oxide	34 (65.4%)	16 (76.2%)	18 (58.1%)	0.17	
Postoperative acute kidney injury requiring renal replacement therapy	14 (26.9%)	11 (52.4%)	3 (9.7%)	0.001*	
Postoperative pneumonia	24 (46.2%)	8 (38.1%)	16 (51.6%)	0.34	
Postoperative septicemia	10 (19.2%)	6 (28.6%)	4 (12.9%)	0.16	
Postoperative fatal arrhythmia	4 (7.7%)	4 (19.0%0	0	0.02*	
Total intensive care unit stay	8.5 (1–167)	5 (1–134)	11 (1–167)	0.59	
Total hospital length of stay	23 (1–206)	12 (1–151)	24 (9–206)	0.99	
Notes.

Data represented by median (range), mean ± SD and number (% within column).

Statistically significant at p-value < 0.05 by Chi-square or Fishers exact test and independent T test (for normally distributed data) or Wilcoxon rank-sum test (for non-normally distributed data) TA, truncus arteriosus.

ECMO extracorporeal membrane oxygenator

Table 3 Characteristics and causes of death following operation (n = 21).

ID	Operation year	Gender	Associated anomalies	Age at repair (days)	Interval post operation (days)	Repaired TA type	Cause of death	
64	2003	Male	None	133	1	Type I	Early postoperative PH crisis	
50	2004	Female	None	47	1	Type II	Early postoperative LCOS, myocardial failure, AKI, possible PH crisis, myocardial ischemia	
62	2004	Female	Microcephaly	92	2	Type I	Early postoperative PH crisis, hemopericardium	
85	2004	Female	Congenital iris cyst	49	14	Type I	Early postoperative PH crisis, sepsis, pneumonia, atrial tachycardia, idioventricular rhythm	
22	2005	Female	None	47	0	Type III with proximal pulmonary artery stenosis	Early postoperative PH crisis	
86	2007	Male	None	135	4	Type III	Early postoperative PH crisis, AKI	
60	2008	Male	DiGeorge syndrome	319	380	Type I	Late mortality due to Persistent PAH post-operation, infection, pneumonia	
46	2009	Female	None	86	1	Type I	Early postoperative myocardial failure possible myocardial ischemia, JET, AV block	
61	2011	Male	None	67	0	Type I, repaired truncal valve	Early postoperative PH crisis, LCOS, VT, VF	
23	2012	Female	Ex preterm, congenital hypothyroid	184	8	Type I	PH crisis, myocardial failure, prolonged CBP on ECMO	
24	2013	Male	Ex preterm, hypospadias	77	459	Type I	Late mortality, persistent PAH post-operation, conduit failure, redo conduit change and died due to pneumonia post reoperation 2 months)	
19	2013	Female	Fetal alcohol syndrome	90	219	Type II	Late mortality: septic shock	
78	2013	Male	None	134	5	Type I, post pulmonary artery banding	Early postoperative PH crisis	
93	2014	Male	Complete bilateral cleft lips and cleft palate	60	25	Type II	Early postoperative PH crisis, LCOS on ECMO, severe intrathoracic infection, bowel ischemia, septic shock	
26	2014	Female	Tracheoesophageal fistula	22	168	Type I	Late mortality: sepsis, pneumonia	
25	2015	Male	None	115	40	Type I	In-hospital mortality, postoperative septic shock, pneumonia	
83	2015	Female	None	251	0	Type II	Early postoperative PH crisis, VT, VF, arrest on ECMO, cardiac tamponade	
31	2015	Female	None	93	1	Type II + repaired IAA, multiple small muscular VSD left opened	Early postoperative PH crisis	
20	2015	Male	DiGeorge syndrome, hypospadia	36	214	Type I	Late mortality: sepsis, pneumonia	
21	2016	Female	DiGeorge syndrome	88	7	Type II	Early postoperative PH crisis	
92	2018	Female	VACTREL association, rib anomaly, hemivertebra	27	4	Type I + repaired truncal valve	Early postoperative PH crisis, sepsis, pneumonia	
Notes.

TA truncus arteriosus

PH crisis pulmonary hypertensive crisis

LCOS low cardiac output syndrome

AKI acute kidney injury

ECMO extracorporeal membrane oxygenator

PAH pulmonary arterial hypertension

VT ventricular tachycardia

VF ventricular fibrillation

IAA interrupted aortic arch

VSD ventricular septal defect

Table 4 Predictors of overall postoperative mortality (n = 52).

Variables	Crude HR (95%CI)	p-value	Adjusted HR (95%CI)	p-value	
Male gender	0.69 (0.29–1.66)	0.42			
Presence of associated anomaly	2.06 (0.87–4.66)	0.09	1.52 (0.59–3.87)	0.38	
DiGeorge syndrome	1.16 (0.39–3.45)	0.78			
Prenatal diagnosis	1.65 (0.38–7.14)	0.49			
Weight at operation <4 kg.	3.18 (1.31–7.71)	0.01*	2.71 (1.05–6.95)	0.039*	
TA type II and III	0.59 (0.25–1.43)	0.25			
Presence of moderate and severe truncal valve regurgitation	2.26 (0.66–7.78)	0.19			
Presence of pulmonary artery stenosis	0.63 (0.18–2.17)	0.47			
Presence of interrupted aortic arch	4.66 (0.59–36.41)	0.14			
Presence of partial anomalous pulmonary venous return	0.04 (0-24.57)	0.33			
Presence of coronary abnormalities	1.61 (0.54–4.81)	0.38			
Pulmonary artery banding prior to repair	0.71 (0.09–5.3)	0.79			
Preoperative major infection within 1 month	2.42 (1.0–6.04)	0.05*	1.23 (0.42–3.61)	0.69	
Preoperative being on mechanical ventilator	1.1 (0.15–8.61)	0.88			
Operation in 1997–2006	1.39 (0.56–3.45)	0.47			
CBP time >150 min	0.45 (0.15–1.32)	0.14			
Intraoperative usage of inhaled nitric oxide	2.24 (0.87–5.71)	0.09	1.71 (0.62–4.71)	0.29	
Notes.

Univariate and multivariate analysis by Cox regression.

*Statistically significant at p-value < 0.05.

HR hazard ratio

TA truncus arteriosus

CBP cardiopulmonary bypass

Of the 22 patients who received palliative treatment, 11 were mortality cases; 7 patients died in infancy (less than 1.5 years of age) due to heart failure. These 7 patients had no indication for receiving PAH therapy. Four of the patients who died (13–32.8 years of age) had late referrals and were diagnosed with ES. Of these four patients, two received oral pulmonary vasodilator (one beraprost and one sildenafil). Of the 11 surviving patients, 6 received oral pulmonary vasodilator (3 beraprost and 3 sildenafil).

Discussion

In this 20-year, single-center database, 52 of 74 patients with a diagnosis of TA had undergone total repair at a median age of 133 days (range: 22 days–16.7 years). Repair of complex TA, which required truncal valve repair or aortic arch interruption repair was performed in 10% of the cases. Early and late mortality was 30.8% and 9.6%, respectively. Mean survival rate of early repaired TA survivors (n = 36) was 91.7% at 1-year and 85.9% at 2-years postoperatively. The independent risk factor for overall mortality was weight at operation <4 kg (hazard ratio: 3.05, 95% CI [1.05–8.74], p-value 0.041). At the median postoperative time of 6.4 years (range: 1.0–19.8 years), more than half of the late-survivors required either catheter intervention or reoperation for conduit change. Freedom from reoperation in repaired TA survivors was 100%, 93%, 70.4%, and 31%, at 1, 5, 10, and 15 years, respectively. All survivors were in the WHO functional class I-II. Of the 22 patients who had palliative treatment (median age at the most recent visit: 13.6 years; range: 14 days–32.8 years), 11 patients (50%) died, with survival rates of 72.7%, 68.2%, 68.2%, and 56.8, at 1, 5, 10, and 15 years of age. This cohort was first to be used in a report of long-term outcomes of patients with TA in Thailand, following the report of Loahaprasittiporn and colleagues who reported early outcomes using the 1995-2004 database from our medical center (Laohaprasitiporn et al., 2008). Survival here is shown with aggregated times and numbers of patients. Although the surgical mortality in this study is still high, it is less than the 50% reported previously for our medical center (Laohaprasitiporn et al., 2008). Furthermore, this cohort includes outcomes for unrepaired TA patients. These findings may aid in counseling patients with regards to treatment and prognosis.

The timing of surgical repair of TA has an impact on outcome as early mortality risk is associated with elevated pulmonary vascular resistance leading to postoperative pulmonary hypertensive crisis (Brown et al., 2001; Hanley et al., 1993; Urban et al., 1998). Hanley and colleagues indicated that when the age at repair was above 100 days, it was an independent predictor of perioperative death. Moreover, pulmonary artery pressure was significantly lower in patients undergoing the operation during the neonatal period (Hanley et al., 1993), which agrees with the study of Urban who found fewer pulmonary hypertensive episodes in patients with repaired TA under 90 days of age (7%), in contrast to above 90 days of age (40%) (Urban et al., 1998). Elective repair during the first three months of age is historically advocated in many centers (Brizard et al., 1997; Lacour-Gayet et al., 1996), though the benefits from repairing TA in the neonatal period is widely accepted (Brown et al., 2001; Hanley et al., 1993; Urban et al., 1998; Naimo et al., 2016). In this decade, the median age at the time of anatomical correction of TA has decreased significantly in many medical centers and has favorably affected surgical outcomes (Naimo et al., 2016; Ivanov et al., 2019). In developing countries; however, delaying the referral often leads to delaying the operation, which remains a health care issue. At a few centers, prenatal diagnosis has gradually developed, though in our study cohort, only four patients had prenatal diagnoses. The median age of referral to our center tended to decrease from 126 days (0–13.9 years of age) in 1995-2006 to 63.5 days (0–25.9 years of age) after 2006 with increased operable opportunity. The age of the patient at time of operation; however, was mostly during infancy (median: 133 days). At our medical center, in 2014, neonatal repairs were initiated and performed in two patients. Unfortunately, one died in hospital and the other had late death at 168 days postoperatively due to pneumonia. In our study, six patients who survived beyond their first year of life had undergone anatomical repair for TA. Pre-operative cardiac catheterization showed that the mean pulmonary artery pressure was 57.2 ± 17.6 mmHg and the pulmonary vascular resistance index in room air, and after a pulmonary vasodilator were 6.2 ± 4.9 and 1.6 ± 1.2 WU m2, respectively. All of these patients survived postoperatively and remained in follow-up until the end of study with a median duration of 5.6 years (range: 4.1–12.7 years). These findings suggest that repair of TA in patients who are older than one year of age is feasible, though physicians need to deliberately select the cases (Arslan et al., 2014; Chen et al., 2016).

A number of single-center and multicenter studies have recently reported acceptable operative outcomes for patients with TA. In-hospital mortality varied from 5 to 17.5% (Brown et al., 2001; Mastropietro et al., 2019; Naimo et al., 2016; Ivanov et al., 2019; Asagai et al., 2016; Morgan et al., 2019), and the presence of associated cardiac anomalies, such as interrupted aortic arch, significant truncal valve regurgitation, coronary abnormalities, and pulmonary arteries were reported to increase the operative mortality (Brown et al., 2001; Hanley et al., 1993; Russell et al., 2012; Naimo et al., 2016). In a large cohort of patients with STS-CHSD, complex TA (defined as TA with significant aortic arch anomaly, interruption or coarctation, and truncal valve regurgitation requiring concomitant repair) accounted for 10% of the TA population, carrying an operative mortality of 30%, which is much higher than that of simple TA repair (6.9–11%) (Russell et al., 2012; Mastropietro et al., 2019). In accordance with a recent report from a Toronto group, the 10-year survival of patients with complex TA (defined as TA with significant aortic arch anomaly, interruption or coarctation, truncal valve regurgitation requiring concomitant repair, and branch pulmonary artery stenosis/ hypoplasia) who had undergone an operation since 2000 was 68%, compared to 95% for patients who had simple TA (Morgan et al., 2019). The overall operative mortality for these patients decreased from 36% to 7% since 2000, which coincided with the average reduction in cardiopulmonary bypass time. Low birth weight, complex TA, and year of diagnosis prior to 2000 were associated with decreased survival for patients up to one year of age in the study by Morgan et al. (2019). In a large study from a Melbourne group, 11.7% early mortality and 11.1% late death were reported following anatomical repair (Naimo et al., 2016). Weight at repair <2.5 kg, prior surgical intervention, coronary abnormalities, and use of postoperative extracorporeal membrane oxygenator were found to be risk factors for early death while DiGeorge syndrome was a risk factor for late death (Naimo et al., 2016). Comparing to our initial report from our medical center (Laohaprasitiporn et al., 2008), early mortality following TA repair was reduced from 50% to 30.8%. This decrease was greater than that recently reported for experienced centers (Russell et al., 2012; Mastropietro et al., 2019; Naimo et al., 2016; Ivanov et al., 2019; Chen et al., 2016; Morgan et al., 2019), but comparable to the 2016 publication from Asagai and colleagues who reviewed 52 patients with repaired TA between 1974 and 2002 (Asagai et al., 2016). The plausible explanation may be because both Asagai’s (Asagai et al., 2016) and ours report had mostly included patients who were beyond the neonatal period. Pulmonary hypertensive crisis and LCOS, followed by precipitated infection were the main causes of early death. The independent risk factor for overall mortality was weight at operation <4 kg (HR 3.05, 95% CI [1.05–8.74], p-value 0.041). Complex TA, coronary abnormality, and extracardiac associated anomalies were not found to be strong predictors of death in our study. The number of truncal valve and interrupted arch repairs in our study was only five, which may have been too small a sample size to demonstrate a statistic difference in the multivariate analysis. The reports of late death (9.6%) were consistent with several other previous studies (Naimo et al., 2016; Ivanov et al., 2019; Asagai et al., 2016; Rajasinghe et al., 1997).

The optimal methods for right ventricular outflow tract reconstruction for TA repair have been long debated (Brown et al., 2001; Lacour-Gayet et al., 1996; Poynter et al., 2013; Hickey et al., 2008). Homograft conduit provides good biocompatibility, stable hemodynamic, and a competent valve to deal with the elevated pulmonary pressure in TA; however, its poor growth potential, especially in TA repair that is usually performed in infancy, leads to an unsatisfactory freedom from reoperation (Poynter et al., 2013). Heterograft valve conduit has been developed and may be used based on material limitations and the desired size of the homograft. Drawbacks have been reported such as possible unfavorable biocompatibility resulting in pseudomembrane formation, aneurysm, and thrombosis and its poor longevity (Hickey et al., 2008). The direct anastomosis technique without extracardiac conduit was proposed by Barbero-Marcial & Tanamati (1999), and it may overcome the limited growth of conduit and decrease the need for reintervention (Raisky et al., 2009), though progressive pulmonary regurgitation, postoperative pulmonary branch stenosis, and compression to the left anterior descending artery, especially in patients with coronary abnormalities, remain as significant issues (Lacour-Gayet et al., 1996; Ivanov et al., 2019). In our medical center, homografts were used in 15 patients (28%) and heterografts were used in 36 patients (69%). Of the 36 heterografts, 24 patients who underwent the operation after 2010 were repaired with a Contegra bovine valve conduit. Conduit size selection in the center was within z-score of 2.3 ± 0.9, which was within the acceptable range mentioned in previous publications (Poynter et al., 2013; Hickey et al., 2008). One patient in this cohort who underwent TA repair with direct anastomosis at 49 days of age died early postoperatively due to pulmonary hypertensive crisis, sepsis, and myocardial failure with atrial tachycardia followed by idioventricular rhythm. In some cases, following TA repair, re-intervention may be performed to conduit and branch the pulmonary arteries (Mavroudis, Jonas & Bove, 2015; Naimo et al., 2016; Ivanov et al., 2019; Morgan et al., 2019; Rajasinghe et al., 1997). Catheter intervention such as balloon dilatation or stent implantation may be initially used at our center in the management to alleviate stenotic lesions as a strategy for delaying reoperation. At the median postoperative duration (6.4 years; range: 1.0–19.8 years), the freedom from re-intervention in the study was 48.3% and 43.9%, at 5 and 10 years, respectively, which is consistent with previous reports  (Russell et al., 2012; Naimo et al., 2016; Ivanov et al., 2019; Morgan et al., 2019). Reoperation for conduit replacement was performed in 8 patients (25.8%) at a median time of 8.7 years (range: 2.7–14.6 years) post total correction. Freedom from reoperation in survivors with repaired TA at 10 years was 70.4%. With regards to truncal valve reoperation, a 14-year-old patient who had undergone concomitant truncal valve repair with primary TA repair, was on a list for reoperation to change the conduit and redo the truncal valve repair at 13 years from the first operation. Among those with a less than moderate truncal valve, 1 patient had infective endocarditis with vegetation at the truncal valve, requiring late truncal valve repair and conduit change at 5.2 years post-operation, and 1 patient had progressive to moderate truncal regurgitation with stable hemodynamic.

Of 22 patients who had TA with palliative treatment, 12 patients were definitely late diagnosed and then referred, which led to a progression of the disease to severe pulmonary arterial hypertension or Eisenmenger syndrome, where total repair should not be performed. Four patients were referred in infancy, and were on the list for TA repair, but they died while waiting for surgery due to infection and multi-organ failure at their local hospital. Three patients were diagnosed and then referred as neonates, but they died from neonatal sepsis/NEC/pneumonia in the center. The parents of two patients denied surgery for their children. One patient had Jacobsen syndrome with severe thrombocytopenia, which has a high risk for major surgery. The parents agreed to have palliative pulmonary artery banding to alleviate heart failure symptoms. The economic burden may play a role in late diagnoses and referrals and in the decision to deny surgery in the earlier era. The fate of patients with palliative treatment for TA was also a focus of this study. Mortality can be classified into 2 groups: the first group (n = 7) died within 1.5 years of age due to intractable heart failure and infection. The second group (n = 4) died at 10-years and older due to ES and progressive truncal valve regurgitation, precipitated by infection. In previous publications, almost 80% of the patients with unrepaired TA died before 1 year of age and few survived until adulthood (Marcelletti, McGoon & Mair, 1976; Niwa et al., 1999; Williams et al., 1999). Niwa and colleagues reviewed 10 adults with ES secondary to unrepaired TA and found the mean age of survival was 41.5 ± 5.1 years, which was a shorter life-span compared to ES secondary to ventricular septal defect (Niwa et al., 1999). In our series, the survival rates of 22 patients with unrepaired TA were 72.7%, 68.2%, 68.2%, and 56.8 at 1, 5, 10, and 15 years of age. Half of these patients were deceased at an estimated 15.2 ± 3.8 years of age (median ± SE). All survivors that were encountered progressed to the ES stage that led to hypoxemia and limited somatic growth. In comparing the ages of survival for patients with repaired and unrepaired TA, no statistical difference was found since most of the operative mortalities included late death occurring before 2 years of age. The survival curve of 36 early survivors following TA repair was superior to that of 22 patients with palliative treatment (Log rank p = 0.03). Therefore, complete repair of TA likely improves the survival of patients with TA. To improve public health outcomes for patients with this lesion, early detection and referral can help manage and repair the lesion in patients who are neonates or at least younger than three months of age. The optimal surgical and postoperative care in neonates and infants in the center is the main focus.

Limitations

Selective bias is inevitable in retrospective studies. Consequently, we included patients with diagnoses of TA type I, II, or III who had undergone TA repair or received palliative treatment at the medical center. All patients were in follow-up or their functional status was known at the end of study (2018). The small number of patients in the study affected the multivariate analysis. In addition, the age of referral and diagnosis of TA at the center is mainly known for patients who were older than neonate, because of the limited resources available in the developing country. Management and surgical strategies also tend to be inconsistent, as they often depend on the preferences of individual surgeons and pediatric cardiologists.

Conclusion

Contemporary survival rates of patients with TA following operation in the center has been gradually improved over time. Most of the operative mortality occurs in the early postoperative period. Compared to patients with TA who had palliative treatment, operative survivors have a better functional status even though they carry a risk for re-intervention. Weight at operation under 4 kg is identified as a significant early and all-mortality risk factor. The survival rate of patients with repaired TA, who can be discharged from hospital after their operation is better than patients with unrepaired TA. We encourage primary physicians to detect this lesion as early as possible for the optimal repair and management.

Supplemental Information

Supplemental Information 1 Raw data of 74 patients with truncus arteriosus

Click here for additional data file.

Supplemental Information 2 Codebook

Click here for additional data file.

The authors thank the faculty and staff of the Cardiovascular Thoracic Surgery, Faculty of Medicine Siriaj Hospital for their support and care of patients with TA. We acknowledge Prof. Duangmanee Loahaprasitiporn, Deputy Dean of Quality Development, Faculty of Medicine Siriaj Hospital, who established the original research and published the preliminary outcomes of patients with TA in Siriraj Hospital since 2008. We also thank Miss Julaporn Poolium who greatly assisted in the statistical analysis.

Additional Information and Declarations

Competing Interests

Author Contributions

Human Ethics

Data Availability

The authors declare there are no competing interests.

Ekkachai Dangrungroj conceived and designed the experiments, performed the experiments, analyzed the data, prepared figures and/or tables, and approved the final draft.

Chodchanok Vijarnsorn conceived and designed the experiments, performed the experiments, analyzed the data, prepared figures and/or tables, authored or reviewed drafts of the paper, and approved the final draft.

Prakul Chanthong, Paweena Chungsomprasong, Supaluck Kanjanauthai, Kritvikrom Durongpisitkul, Jarupim Soongswang, Kriangkrai Tantiwongkosri, Thaworn Subtaweesin and Somchai Sriyoschati performed the experiments, authored or reviewed drafts of the paper, and approved the final draft.

The following information was supplied relating to ethical approvals (i.e., approving body and any reference numbers):

The Siriraj Institutional Review Board Faculty of Medicine, Siriraj Hospital, Mahidol University has approved this study (COA no. Si 379/2017).

The following information was supplied regarding data availability:

The raw data set is available as a Supplemental File.

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
