# Peer review of "Long-term outcomes of repaired and unrepaired truncus arteriosus: 20-year, single-center experience in Thailand"

_PeerJ, doi:10.7717/peerj.9148_

## Round 0.1 · original submission · Major Revisions

The MS needs deep revision following indication of reviewers

Reviewer 1's report is in their attached PDF

Reviewer 1 ·

Basic reporting

no comment

Experimental design

no comment

Validity of the findings

no comment

Additional comments

no comment

Annotated reviews are not available for download in order to protect the identity of reviewers who chose to remain anonymous.

Reviewer 2 ·

Basic reporting

In this paper Dangrungroi et al. presented their 20 year experience with truncus arteriosus (TA) repair and also long-term outcome of the unrepaired patients. This is an outcome study. The authors could be congratulated that the hospital mortality has dropped from the 50 % (according to their previous reports) to the 30.8 % in the current era.
The paper was written in clear academic English, it is well organised in terms of reading it through. The references are also well organised and are relevant. Most of the figures and tables are well structured, the only issue that I have found was in the Table 4. Mainly, they should have included the number of patients in each category (e.g prenatal diagnosis in 20 out of 52 patients). There are also some minor errors regarding presenting data (e.g.Table 2; Surgical era 1999-2006 instead of operation in 1999-2006, however they do not distort meaning. The raw data that authors collected is robust.

Experimental design

In terms of experimental design, the finding of the paper is not completely in line with its aims. In abstract the authors stated that their aim was as follow: “In this report, we aim to report on the survival of patients with TA in our medical center in the recent era”(lines 35-36). However, in conclusion they reported on functional status and then on decreased mortality risk (lines 54-55). I suspect they wanted to say about early mortality as they define previously (lines 114-116), but not on mortality risks. Also, they did not aim to report on risk factors, though they reported on them in conclusion (line 56).
I also suspect the multivariate analysis by Cox regression is meaningless, as the authors did not provide the number of events in each category (Table 4), and also considering overall small group (n=52), the number of events could not be more than 10 in each category.

Validity of the findings

With regards to novelty in the field of cardiac surgery this paper does not contribute much. The main problems are as follows:
1. The quoted “crude-for simple and complex TA” hospital mortality of 30.8 %. However, the current hospital mortality is within 5 % and 17.5 % (line 270). Moreover, the complex TA accounted to 9.6 % (5/52) of patients, so such high number of hospital mortality are not in line with worldwide numbers. The authors also have not provided the hospital mortality for the complex TA.
2. The comparison of survival between operated and non-operated groups is without sense (lines 213-215). By definition, unrepaired TA leads to early death (most of unrepaired patients died by one year of life( line 342) and those who “survive” are profoundly disable due to Eisenmenger complex. Relatively old age in the group of unrepaired patients could be explained by lack of early diagnosis (Table 1; the age of suspicion of 0.07 years versus the age of admission to the cardiac center of 1.8 years in the group of unrepaired patients. That means that most of the patients were lost and probably died.

---

## Round 0.2 · Minor Revisions

There are some further minor comments to be addressed. Please provide feedback accordingly.

Reviewer 1 ·

Basic reporting

I think the revision stratified the previous comments. The manuscript can be published with some minor revision.

Experimental design

NO COMMENTS

Validity of the findings

NO COMMENTS

Additional comments

a Please delete the description for the definition of TA in the Introduction section,
b Is Siriraj Hospital the only heart center which can offer the service for the patients with TA in Thailand? If not, please delete the first sentence in the 2nd paragraph.
c Please indicate whether the targeted advance PAH therapy is utilized for the TA patients with or without surgical treatment, because such a therapy could improve the prognosis of the patients with PAH.

---

## Round 0.3 · accepted · Accept

The authors significantly improved the manuscript after revision. It can now be accepted.